# A Convolutional Dynamic-Jerk-Planning Algorithm for Impedance Control of Variable-Stiffness Cable-Driven Manipulators

**DOI:** 10.3390/mi13112021

**Published:** 2022-11-19

**Authors:** Luyang Zhang, Lihui Jia, Panpan Yang, Zixuan Li, Yuhuan Zhang, Xiang Cheng, Zonggao Mu

**Affiliations:** Shandong Provincial Key Laboratory of Precision Manufacturing and Non-Traditional Machining, School of Mechanical Engineering, Shandong University of Technology, Zibo 255000, China

**Keywords:** cable-driven manipulators, motion control, impedance control, variable-stiffness

## Abstract

Cable-driven manipulators, characterized by slender arms, dexterous motion, and controllable stiffness, have great prospects for application to capture on-orbit satellites. However, it is difficult to achieve effective motion planning and stiffness control of cable-driven manipulators because of the coupled relationships between cable lengths, joint angles, and reaction forces. Therefore, a convolutional dynamic-jerk-planning algorithm is devised for impedance control of variable-stiffness cable-driven manipulators. First, a variable-stiffness cable-driven manipulator with universal modules and rotary quick-change modules is designed to overcome difficulties related to disassembly, installation, and maintenance. Second, a convolutional dynamic-jerk-planning algorithm is devised to overcome the discontinuity and shock problems of the manipulator’s velocity during intermittent control processes. The algorithm can also make acceleration smooth by setting jerk dynamically, reducing acceleration shock and ensuring the stable movement of the cable-driven manipulator. Third, the stiffness of the cable-driven manipulator is further optimized by compensating for the position and velocity of drive cables by employing position-based impedance control. Finally, the prototype of the variable-stiffness cable-driven manipulator is developed and tested. The convolutional dynamic-jerk-planning algorithm is used to plan the desired velocity curves for velocity control experiments of the cable-driven manipulator. The results verify that the algorithm can improve the acceleration smoothness, thereby making movement smooth and reducing vibrations. Furthermore, stiffness control experiments verify that the cable-driven manipulator has ideal variable stiffness capabilities.

## 1. Introduction

With the rapid development of space technology, various satellites have played important roles in the navigation, meteorology, and other fields. However, the failure of high-value satellites will bring huge losses. Since the above satellites are often non-cooperative, the corresponding on-orbit capture technology has attracted great attention [1,2,3,4,5]. Among them, the capturing of on-orbit satellites without pre-designed docking mechanisms is still an urgent difficulty to be solved [6,7]. Due to the rigid impact caused by the capture collision, traditional 6–7 degrees of freedom (DOF) manipulators composed of rigid joints can only perform capture tasks of cooperative targets in structured environments [8]. Fortunately, cable-driven manipulators [9,10,11] with characteristics of dexterous motion and controllable stiffness have demonstrated great potential for compliant capture of non-cooperative satellites [12,13]. Therefore, according to requirements of compliant capture of satellites, the dexterous motion control [14] and variable stiffness control [15] of cable-driven manipulators have become the focus of current research. In terms of motion control, the iterative Jacobian method [16,17] is usually used to control the motion of the driving cable. This method can only control the velocity by the error between the desired length and the actual length, so it will cause the velocity to become discontinuous. In order to solve the discontinuity problem of velocity, Tang [18] proposed a two-level motion planning method. The method plans a continuous velocity curve to improve the motion smoothness and stability of the cable-driven manipulator. However, this method is tedious and computationally inefficient through the process of polynomial planning of time, velocity, and acceleration. The setting of acceleration and jerk are inflexible and non-universal. Therefore, this paper proposes a convolutional dynamic-jerk-planning algorithm. The algorithm cannot only optimize the velocity curve and improve the position accuracy of the cable-driven manipulator, but also simplify the calculation efficiency and improve the operation flexibility. In terms of stiffness control, Nan Ma et al. [19] analyzed the advantages and disadvantages of various 2-DOF rotating mechanisms. A 2-DOF cable-driven parallel mechanism with simple processing and adjustable stiffness was designed. Then, a stiffness model of the mechanism was established. Compared with experiments, the total deviation of the model was very small, which could be used as a reference for the study of joint stiffness models. In order to solve the problem of low stiffness, Liu [20] designed a new mechanism to improve the stiffness performance and load capacity of the cable-driven manipulator. This mechanism also improved the position accuracy of the cable-driven manipulator. By redundantly arranging elastic joints and constraining the bending curvature with curvature restraining rods, Zhao et al. [21] improved the stiffness of the continuum manipulator and derived its stiffness formula. Zhang et al. [22] analyzed the influencing factors of stiffness by modeling the end of the continuum manipulator. A control method using flexible struts and segmented driving manipulators was proposed to control the deformation of the continuum manipulators. According to the influence of the length distribution of the connecting rod on the workspace, Wu [23] established a stiffness optimization model to provide the research basis for the design and variable stiffness control of the cable-driven manipulator. On the basis of studying the stiffness model, many scholars have adopted the method of changing the material and designing a new mechanism to improve the stiffness of cable-driven manipulators. However, there are few studies on the motion control algorithm and stiffness control algorithm of the cable-driven manipulator [24].

In order to study the motion planning and stiffness control of the cable-driven manipulator, a variable-stiffness cable-driven manipulator with rotary quick-change module is designed and experimented on in this paper. The rest of the paper is organized as follows. Section 2 briefly introduces the structure design and the analysis of variable-stiffness cable-driven manipulators. Then, the kinematics model is detailed. In order to overcome the discontinuity and shock problem of velocity, the convolutional dynamic-jerk-planning algorithm is proposed in Section 3. The position-based impedance control algorithm of the variable-stiffness cable-driven manipulator is detailed in Section 4. In Section 5, the prototype of the variable-stiffness cable-driven manipulator is introduced and is subjected to experiments. Section 6 presents the conclusion of this paper.

## 2. Structure of Variable-Stiffness Cable-Driven Manipulators

Variable-stiffness cable-driven manipulators have great promise for use in capturing non-cooperative satellites, as depicted in Figure 1. These manipulators have three defining characteristics:
Separate Layout of Mechanical and Electrical Components. The motors, controllers, and other electrical components of a manipulator are uniformly installed on a highly protected satellite platform, which enhances their reliability in harsh environments. This design also means that a manipulator is light weight and has low inertia, such that its motion has little effect on its satellite platform.Organic Integration of Rigid and Flexible Components. A manipulator achieves a rigid and flexible variable-stiffness working effect by controlling its configuration and cable tension. Thus, a manipulator is suited for use in the low-impact capture and high-stiffness manipulation of non-cooperative satellites.Dexterous Motion. A manipulator has both hyper-redundant DOF and slender linkages. Thus, it can move dexterously in unconstrained and narrow environments with many obstacles and perform maintenance tasks in complex environments.

The variable-stiffness cable-driven manipulator studied herein has a modular design, with a universal module consisting of a universal joint and a fixed linkage, as shown in Figure 2a. Each universal module is controlled by three drive cables evenly distributed along the linkage. The motion of each drive cable is controlled by a motor-driven screw. The manipulator consists of four universal modules connected in series in a “yaw-pitch—pitch-yaw—yaw-pitch—pitch-yaw” configuration, as shown in Figure 2a. Overall, the manipulator has eight DOFs and 12 drive cables, with each universal module having two DOFs controlled by three drive cables.

### 2.1. Structure Design

To achieve rapid disassembly, installation, and maintenance with a stable and simple mechanical structure, this paper designs an innovative variable-stiffness cable-driven manipulator with a drive system based on a rotary quick-change module. This drive system comprises six rotary quick-change modules and one central shaft. The central shaft constrains the six rotary quick-change modules between the two fixed plates, as shown in Figure 2b. After the two ends of the fixed shaft are released, the rotary quick-change module can freely rotate around the central shaft of the drive system to achieve rapid disassembly, installation, and maintenance. The rotary quick-change module comprises screws, junction plates, T-nuts, fixed plates, drive cables, sliders, limiters, couplings, and motors (i.e., RM2006), as shown in Figure 2c. Two sets of motors and screws are locked onto the central shaft by two fixed shafts. Each motor controls one corresponding drive cable. The motor is connected to the screw by the coupling. The rotary motion of the lead screw leads to the translational motion of the T-nut. The T-nut thus moves the slider along the central shaft to change the length of the drive cable. One end of the drive cable is fixed to the universal module, while the other end is fixed to the tension sensor through the limiter. The maximum radial load produced by the motor is 495 N. Therefore, the cable-driven manipulator designed in this paper demonstrates the potential characteristics of variable stiffness.

### 2.2. Kinematic Model

In order to achieve dexterous movement in narrow environments with many obstacles, the kinematic models of the variable-stiffness cable-driven manipulator among joint space, actuator space, and task space are derived in this section.

The kinematic model is established to describe the relative kinematic relationship between joint space and task space. First, according to the structure of the variable-stiffness cable-driven manipulator, the Denavit–Hartenberg (D-H) coordinate system is established in Table 1 and Figure 3.

It can be known that the homogeneous transformation matrix between two adjacent coordinate system is as (1).
(1)Tii−1=[cosθi−sinθicosαisinθisinαiaicosθisinθicosθicosαi−cosθisinαiaisinθi0sαicαidi0001]

Then, the forward kinematics between joint space and task space of the variable-stiffness cable-driven manipulator can be written as (2).
(2)Tn0=T10T21⋯⋯Tnn−1(n=1,2,3⋯7,8)

## 3. Convolutional Dynamic-Jerk-Planning Algorithm for Variable-Stiffness Cable-Driven Manipulators

### 3.1. Principle of Convolutional Dynamic-Jerk-Planning Algorithm

The principle of the convolutional dynamic-jerk-planning algorithm used to control the cable-driven manipulators studied herein is detailed in Figure 4. This algorithm is based on the characteristic of digital convolution, where the total area of the input and output function curves of a convolution operation remains unchanged. The function settings of this convolution process are described as follows. The convolution operator $h_{a}\left(t\right)$ of the first convolutional dynamic-jerk-planning algorithm is expressed in Equation (3), as follows:
(3)$$h_{a}(t)=\frac{1}{t_{a}}$$
where $h_{a}\left(t\right)$ has the domain $\left(0,t_{a}\right)$, and $t_{a}$ is the first convolution acceleration time. The input function of the first convolution is set by Equation (4), as follows: (4)$$y_{0}\left(t\right)=v_{0}$$
where $y_{0}\left(t\right)$ and $v_{0}$ are rectangular velocity input functions that determine the desired displacement and velocity of the drive cable. The domain of $y_{0}\left(t\right)$ is $\left(0,t_{0}\right)$, where $t_{0}$ is the time when uniform motion reaches the desired displacement, D. The first convolution of $h_{a}\left(t\right)$ and $y_{0}\left(t\right)$ forms the output function (Equation (5)): (5)$$y_{1}\left(t\right)=\int_{0}^{t_{a}}y_{0}\left(\tau \right)h\left(\tau -t_{a}\right)d\tau$$
where $t_{a}$ is defined by Equation (6): (6)$$t_{a}=v_{0}/a_{\max}$$

In Equations (1)–(4), $h_{a}\left(t\right)$ performs the first convolution operation, which yields a trapezoidal velocity curve $y_{1}\left(t\right)$. However, the acceleration and deceleration states of this trapezoidal velocity curve would cause excessive velocity shock during manipulation. Therefore, $y_{1}\left(t\right)$ is used as the new input function and $h_{j}\left(t\right)$ as the convolution operator to perform a second convolution operation, which produces the velocity curve $y_{2}\left(t\right)$, as follows: (7)$$h_{j}(t)=\frac{1}{t_{j}}$$
(8)$$y_{2}\left(t\right)=\int_{0}^{t_{j}}y_{1}\left(\tau \right)h\left(\tau -t_{j}\right)d\tau$$
where $t_{j}$ is the second convolution acceleration time in Equation (9): (9)$$t_{j}=a_{\max}/j_{\max}$$
where $a_{\max}$ is the maximum acceleration of the uniform acceleration stage, and $j_{\max}$ is the maximum jerk of the variable acceleration stage. This curve now smoothly connects the stationary, acceleration, constant velocity, and deceleration states by controlling the acceleration and jerk. The acceleration and deceleration states both contain variable and uniform acceleration stages, while the acceleration and jerk in this convolution process can both be set freely.

As a demonstration, the above convolution method and an iterative Jacobian method are each used to plan the velocity curve for the movement of the manipulator, with the results shown in Figure 5. The results for the Jacobian method (Figure 5a) show that the acceleration suddenly increases and decreases at the initial and end motion phases, respectively. These rapid movements would shock and vibrate the manipulator, while also degrading its tracking performance and shortening its service life. By contrast, by using our algorithm, the acceleration and jerk in Figure 5b rise and fall steadily at the initial and end motion phases, respectively. This smoother motion would produce smaller system vibrations and impulses, illustrating how the convolutional dynamic-jerk-planning algorithm could reduce acceleration shock and stabilize the manipulator’s motion.

According to the actual operation of the variable−stiffness cable−driven manipulator and the above convolution operation process, the recursive expression of the convolution is determined by Equation (10), as follows: (10)$$y_{n}\left[k\right]=\frac{y_{n-1}\left[k\right]-y_{n-1}\left[k-m_{n}\right]}{m_{n}}+y_{n}\left[k-1\right]$$
where k is the number of current samples and $m_{n}$ is the number of samples corresponding to the $n^{\mathrm{th}}$ convolution.

It can be seen from Equation (10) that the recursive calculation of the convolution is only composed of addition, subtraction, and division operations. It greatly simplifies the calculation process.

The convolutional dynamic-jerk-planning algorithm is simpler to compute than the polynomial-based velocity planning method [19]. The movement of the cable-driven manipulator process has multiple stages, such as variable acceleration, uniform acceleration, uniform speed, and uniform deceleration stages. When in processes with long run times and complex stages, the polynomial-based method requires a large number of equations and parameters to be set. However, the convolution method requires fewer parameters and is simpler to calculate.

### 3.2. Applications of the Convolutional Dynamic-Jerk-Planning Algorithm

According to the principle described in Section 3.1, the convolutional dynamic-jerk-planning algorithm can be modified for different desired displacements. Thus, the midpoint between the planned $y_{I}\left(t\right)$ and $y_{\mathrm{II}}\left(t\right)$ is calculated to produce a new asymmetric velocity curve, as shown in Figure 6. This new curve is characterized by the following set of time points.

$t_{aI}$ in $y_{I}\left(t\right)$, which is the first convolution acceleration time and is given by: (11)$$t_{aI}=v_{\max}/a_{\max}$$
$t_{jI}$ in $y_{I}\left(t\right)$, which is the second convolution acceleration time and is given by: (12)$$t_{jI}=a_{\max}/j_{\max}$$
$t_{a\mathrm{II}}$ in $y_{\mathrm{II}}\left(t\right)$, which is the first convolution acceleration time and is given by: (13)$$t_{a\mathrm{II}}=K_{a}v_{\max}/a_{\max}$$
$t_{j\mathrm{II}}$ in $y_{\mathrm{II}}\left(t\right)$, which is the second convolution acceleration time and is given by: (14)$$t_{j\mathrm{II}}=K_{j}a_{\max}/\left(K_{a}j_{\max}\right)$$
where $v_{\max}$ is the maximum velocity during acceleration, and $K_{a}$ and $K_{j}$ are coefficients.

With this asymmetric velocity curve, the acceleration and jerk of the acceleration and deceleration stages can be set separately, allowing the velocity curve to adapt more flexibly to different displacements. Four displacement situations are defined: ultra-long displacement, long displacement, medium displacement, and short displacement. These different situations are described below.

1.Ultra-long displacement:

When the desired displacement $D>2\max\left(D_{I},D_{\mathrm{II}}\right)$, then $y_{I}\left(t\right)$ and $y_{\mathrm{II}}\left(t\right)$ have uniform motion phases, as shown in Figure 6a. $y_{I}\left(t\right)$ and $y_{\mathrm{II}}\left(t\right)$ are each divided into two equal parts, with their midpoint times expressed by $t_{I}$ and $t_{\mathrm{II}}$: (15)$$t_{I}=\frac{t_{1}+t_{aI}+t_{jI}}{2}$$
(16)$$t_{\mathrm{II}}=\frac{t_{2}+t_{a\mathrm{II}}+t_{j\mathrm{II}}}{2}$$

According to the values calculated in Equations (15) and (16), the interval time $t_{T}$ can be obtained from Equation (17), as follows: (17)$$t_{T}=t_{\mathrm{II}}-t_{I}$$

By superimposing $y_{I}\left(t\right)$ and $y_{\mathrm{II}}\left(t\right)$ at the midpoint, the desired asymmetric velocity curve can be obtained, as shown in Figure 6a. The superimposed function is expressed by Equation (18), as follows: (18)$$y_{d}\left(t\right)=\{\begin{array}{c} \begin{array}{cc} y_{I}\left(t\right), & 0\leq t\leq t_{I} \end{array}\\ \begin{array}{cc} y_{\mathrm{II}}\left(t+t_{T}\right), & t_{I}\leq t\leq t_{I}+t_{\mathrm{II}} \end{array} \end{array}$$

To simplify the calculation process and facilitate practical applications, the recursive expression of $y_{d}\left(t\right)$ can be rewritten, as follows (Equation (19)): (19)$$y_{n}\left[k\right]=\{\begin{array}{c} \frac{y_{\left(n-1\right)I}\left[k\right]-y_{\left(n-1\right)I}\left[k-m_{nI}\right]}{m_{nI}}+y_{nI}\left[k-1\right]\\ \frac{y_{\left(n-1\right)\mathrm{II}}\left[k+k_{T}\right]-y_{\left(n-1\right)\mathrm{II}}\left[k+k_{T}-m_{n\mathrm{II}}\right]}{m_{n\mathrm{II}}}+y_{n\mathrm{II}}\left[k-1\right] \end{array}$$
where n is the number of calculations, k is the number of current samples, $m_{nI}=t_{jI}/t_{s}$,$m_{n\mathrm{II}}=t_{j\mathrm{II}}/t_{s}$, $k_{T}=t_{T}/t_{s}$, and $t_{s}$ is the sampling period.

2.Long displacement:

When $D_{I}+D_{\mathrm{II}}<D\leq 2\max\left(D_{I},D_{\mathrm{II}}\right)$, there is only one uniform motion phase in either $y_{I}\left(t\right)$ or $y_{\mathrm{II}}\left(t\right)$, as shown in Figure 6b. $t_{I1}$ and $t_{\mathrm{II}2}$ are thus defined by Equation (20), as follows: (20)$$\{\begin{array}{c} t_{I}=\left(t_{1}+t_{aI}+t_{jI}\right)/2\\ t_{\mathrm{II}}=t_{a\mathrm{II}}+t_{j\mathrm{II}} \end{array}$$

Therefore, the recursive expression for the long displacement can be derived using Equations (17)–(19).

3.Medium displacement:

When $D_{a}<D\leq D_{I}+D_{I}{}_{I}$, then $y_{I}\left(t\right)$ and $y_{\mathrm{II}}\left(t\right)$ have no uniform motion phase, as shown in Figure 6c,d. However, $y_{I}\left(t\right)$ and $y_{\mathrm{II}}\left(t\right)$ do have uniform acceleration phases, and thus the maximum velocity $a_{\max}$ can be reached at this time. The expressions for $D_{a}$ and $v_{a}$ are as follows (Equations (21) and (22)): (21)$$D_{a}=D_{I_{a}}+D_{{\mathrm{II}}_{a}}=\frac{v_{a}\left(t_{aI}+t_{jI}\right)+v_{a}\left(t_{a\mathrm{II}}+t_{j\mathrm{II}}\right)}{2}$$
(22)$$v_{a}=\frac{a_{\max}^{2}}{j_{\max}}$$

Thus, $t_{I}$ and $t_{\mathrm{II}}$ can be calculated using Equations (23): (23)$$\{\begin{array}{c} t_{I}=t_{aI}+t_{jI}=v_{a}/a_{\max}+t_{jI}\\ t_{\mathrm{II}}=t_{a\mathrm{II}}+t_{j\mathrm{II}} \end{array}$$

Then, using Equations (17)–(19), the recursive expression for the medium displacement can be derived.

4.Short displacement:

When $0<D\leq D_{a}$, then $y_{I}\left(t\right)$ and $y_{\mathrm{II}}\left(t\right)$ have no uniform acceleration phase, as shown in Figure 6d. The current desired displacement D is thus described by the following equations (Equations (24)–(26)): (24)$$D=\frac{v_{I}\left(t_{aI}+t_{jI}\right)+v_{\mathrm{II}}\left(t_{a\mathrm{II}}+t_{j\mathrm{II}}\right)}{2}$$
(25)$$v=\frac{a^{2}}{j_{\max}}$$
(26)$$a=\sqrt[{3}]{\frac{2Dj_{\max}^{2}}{2+K_{a}+K_{j}/K_{a}}}$$

Therefore, $t_{I}$ and $t_{\mathrm{II}}$ are given by Equation (25), as follows: (27)$$\{\begin{array}{c} t_{I}=t_{aI}+t_{jI}=v/a+a/j_{\max}\\ t_{\mathrm{II}}=t_{a\mathrm{II}}+t_{j\mathrm{II}}=K_{a}v/a+K_{j}a/\left(K_{a}j_{\max}\right) \end{array}$$

The recursive expression for the short displacement can then be derived from Equations (17)–(19).

These examples demonstrate how the convolutional dynamic-jerk-planning algorithm can adapt to the cases of ultra-long, long, medium, and short displacements, and also adapt to cases with asymmetric acceleration and deceleration stages. It achieves this adaptability by freely adjusting the magnitude of the acceleration and jerk to suit different displacements of motion.

## 4. Position-Based Impedance Control of Variable-Stiffness Cable-Driven Manipulators

The position-based impedance control is based on the second order spring–mass–damper system and adjusts the tension of the drive cables to control their stiffness. The contact force between the variable-stiffness cable-driven manipulator and the environment can be controlled by adjusting the position, velocity, acceleration, and tension of these drive cables. The impedance relationship model expression is established in Equation (28), as follows: (28)$$M_{d}[\ddot{X}(t)-{\ddot{X}}_{d}(t)]+B_{d}[\dot{X}(t)-{\dot{X}}_{d}(t)]+K_{d}[X(t)-X_{d}(t)]=F_{e}(t)-F_{d}(t)$$
where $X_{d}$, ${\dot{X}}_{d}$, and ${\ddot{X}}_{d}$ are the desired position, velocity, and acceleration of the drive cable, respectively. X, $\dot{X}$, and $\ddot{X}$ are the actual position, velocity, and acceleration of the drive cable, respectively. $F_{e}$ and $F_{d}$ are the actual and desired tension vectors of the drive cables. Additionally, inertia coefficient matrix $M_{d}$, damping coefficient matrix $B_{d}$, and stiffness coefficient matrix $K_{d}$ are impedance control parameters of the variable-stiffness cable-driven manipulator.

The position-based impedance control system is composed of an inner position loop and an outer tension loop. This system controls the position and velocity of the drive cables by adjusting their tension. This system is combined with the actual cases and established in Figure 7.

In the outer tension loop, the relationship between the position compensation of the drive cable and the tension of the drive cable can be expressed as H(s): (29)$$H(s)=\frac{E(s)}{\Delta F(s)}=\frac{1}{M_{d}s^{2}+B_{d}s+K_{d}}$$
(30)$$E\left(s\right)=X\left(s\right)-X_{d}\left(s\right)$$
where H(s) is the admittance characteristic in the frequency domain and E(s) is the difference between the desired and actual position vectors in the frequency domain. By using the bilinear variation $H(z)=H(s)|{}_{s=\frac{2}{T}\frac{z-1}{z+1}}$, the discrete expression of the impedance model can be obtained from Equation (31), as follows: (31)$$H(z)=\frac{E(z)}{\Delta F(z)}=\frac{T^{2}{\left(z+1\right)}^{2}}{w_{1}z^{2}+w_{2}z+w_{3}}$$
(32)$$\{\begin{array}{c} w_{1}=4M_{d}+2B_{d}T+K_{d}T^{2}\\ w_{2}=2K_{d}T^{2}-8M_{d}\\ w_{3}=K_{d}T^{2}+4M_{d}-2B_{d}T \end{array}$$

Equation (31) indicates that the differential equation of the position-based impedance control is defined by Equation (33), as follows: (33)$$e(n)=\frac{1}{w_{1}}\left[\begin{array}{l} \Delta F(n)T^{2}+2T^{2}\Delta F(n-1)+\ldots \\ T^{2}\Delta F(n-2)-w_{2}e(n-1)-w_{3}e(n-2) \end{array}\right]$$
where e is the compensation value and $X_{c}$ is the compensated position vector, which can be obtained from Equation (34), as follows: (34)$$X_{c}=X_{d}+e$$

The actual and compensated angles of the motor are $\theta_{M}$ and $\theta_{c}$, respectively; the actual and compensated angular velocities of the motor are $\theta_{\mathrm{Mv}}$ and $\theta_{\mathrm{vc}}$, respectively; and the lead of the screw is $l_{s}$ = 1 mm. According to the transmission relationship, the changes between the position of the drive cable and the angle of the motor can be calculated using Equation (35): (35)$$X=\frac{\theta_{M}}{2\pi }l_{s}$$

The mapping relationship between ${\dot{\theta }}_{\mathrm{Mv}}\left(r/m\right)$ and v(mm/s), the actual velocity of the drive cable, is described by Equation (36), as follows: (36)$$v=\frac{\theta_{\mathrm{Mv}}}{60\times 2\pi }l_{s}$$

If the error between the desired and actual tension of the drive cable is zero, then the control system can be simplified to the position control of the drive cable. If the error between the desired tension and the actual tension of the drive cable is not zero, then the control system can generate compensation based on the error between the desired tension and the actual tension of the drive cable to correct the desired position.
(37)$$v_{c}=v_{d}+v_{e}$$

In this paper, the inner velocity loop is added, as expressed in Equation (37), to provide position-based impedance control. Here, $v_{e}$ is the derivative of e, $v_{d}$ is the desired velocity calculated by convolutional dynamic-jerk-planning algorithm, and $v_{c}$ is the updated desired velocity of the drive cable after compensation.

## 5. Prototypes and Experiments

The prototype of the variable-stiffness cable-driven manipulator consists of hardware control and software monitoring components, which were integrated into the design of the manipulator system shown in Figure 8. This prototype consists of four universal modules with eight DOFs. The experimental system comprises the prototype, a power supply, a controller, a transducer, a global camera, and a personal computer.

### 5.1. Experiment Setup

Variable stiffness control is particularly important for the flexible capture of satellites. The stiffness of the cable-driven manipulator depends on the configuration of the moving manipulator and the tension in the drive cables. To assess these functionalities of the prototype, a series of velocity control and stiffness control experiments were performed. The entire experiment process was recorded by the global camera, which was fixed to the end of the prototype.

### 5.2. Velocity Control Experiments

The actual velocities of the 12 drive cables should be based on the velocity planned by the convolutional dynamic-jerk-planning algorithm. The velocity mapping between the motor and the drive cable is defined by Equation (35). The actual velocity and acceleration of the 12 drive cables are shown in Figure 9a,b.

The velocity errors of the 12 drive cables were calculated according to the desired velocity in Figure 5b and the actual velocity in Figure 9a. As shown in Figure 9c–f, the velocity error during acceleration and deceleration was ±0.03 mm/s, whereas the velocity error at constant velocity was ±0.01 mm/s. During acceleration and deceleration, the acceleration increases or decreases gradually instead of changing abruptly. These experimental data demonstrate that the convolutional dynamic-jerk-planning algorithm reduces the system vibration caused by excessive acceleration at the beginning and end of the motion. Therefore, applying the algorithm would reduce the acceleration shock and make the manipulator’s movements more stable.

### 5.3. Stiffness Control Experiments

The stiffness of the cable-driven manipulator refers to the ability of the end of the manipulator to resist deformation when it is affected by external forces. Generally, the greater the stiffness of the manipulator, the higher the positioning accuracy of its end. To verify the stiffness control ability of the manipulator, dynamic stiffness experiments were performed. The position-based impedance control system was applied in the dynamic stiffness experiments to control cable tension. The experiments also applied the convolutional dynamic-jerk-planning algorithm to control the displacement and velocity of the cable.

Before the stiffness control experiment, the load experiment of the cable-driven manipulator was conducted in this paper. The cable-driven manipulator was placed on a fixed plane. A weight of 100 g, 200 g, 300 g, and 500 g is applied to the end of the cable-driven manipulator to test its loading capacity. The experimental results are shown in Figure 10. The limit load of the cable-driven manipulator is 500 g.

First, the dynamic stiffness control of the manipulator was established by determining the required displacement and velocity of the drive cables. The manipulator was directed to bend upward, with its motion recorded by the global camera. Then, the manipulator was moved back to its original horizontal state. Next, the end of the manipulator was loaded with a 100 g weight, and the actions of bending upward and returning to the horizontal state were repeated. Finally, the previous actions with the loaded weight were repeated with position-based impedance control applied. The end points of the manipulator’s different trajectories are plotted in Figure 11a, with the full trajectories shown in Figure 11b. Trajectory 1 is the normal motion of the manipulator, trajectory 2 is the manipulator’s motion when loaded with a 100 g weight, and trajectory 3 is the manipulator’s position-based impedance-controlled motion when loaded with a 100 g weight. Trajectory 1 is the standard for movement. Error 1 is the position error between trajectory 1 and trajectory 2, while error 2 is the position error between trajectory 1 and trajectory 3; the corresponding position error curves are shown in Figure 11c. Stiffness 1 is the stiffness of the end of the cable-driven manipulator as it moves along trajectory 2. Stiffness 2 is the stiffness of the end of the cable-driven manipulator as it moves along trajectory 3. The stiffness can be calculated by (38): (38)$$K=\frac{F}{\Delta D}$$
where F is the external force on the end of the cable-driven manipulator in the vertical direction, and ΔD is the change in displacement of the end of the cable-driven manipulator in the vertical direction.

The results showed that the cable-driven manipulator moved smoothly with high precision when not under load in trajectory 1. The manipulator also moved normally when loaded with a 100 g weight but exhibited a large position error compared with trajectory 1. When the final stationary state is reached, the end position error is 16.81 mm and the stiffness value is 59.49 N/m. However, this poor dynamic stiffness performance was significantly improved by applying the position-based impedance control. The position error between trajectory 1 and trajectory 3 is 1.96 mm. The stiffness value is 510.20 N/m. The stiffness of the cable-driven manipulator has been significantly improved. Therefore, the position-based impedance control improved the load capacity and control accuracy of the manipulator.

## 6. Conclusions

A convolutional dynamic-jerk-planning algorithm was devised for impedance control of a variable-stiffness cable-driven manipulator. The manipulator was designed with six rotary quick-change modules and four universal modules to overcome the difficulties of disassembly, installation, and maintenance. The convolutional dynamic-jerk-planning algorithm solved the discontinuity and shock problems of the manipulator’s velocity during the intermittent control process. The algorithm also reduced acceleration shock and made movement more stable by setting jerk dynamically.

Velocity control experiments revealed that this algorithm limited the velocity error during acceleration and deceleration to ±0.03 mm/s and the velocity error at constant velocity to ± 0.01 mm/s. By applying position-based impedance control to compensate for the displacement and velocity of the drive cables, the stiffness of the cable-driven manipulator was further optimized. Stiffness control experiments indicated that applying this control reduced the position error of a loaded manipulator from 16.81 mm to 1.96 mm.

Future studies should investigate the payload capability, stiffness modeling, and force-compliant control of variable-stiffness cable-driven manipulators.

## Figures and Tables

**Figure 1 micromachines-13-02021-f001:**
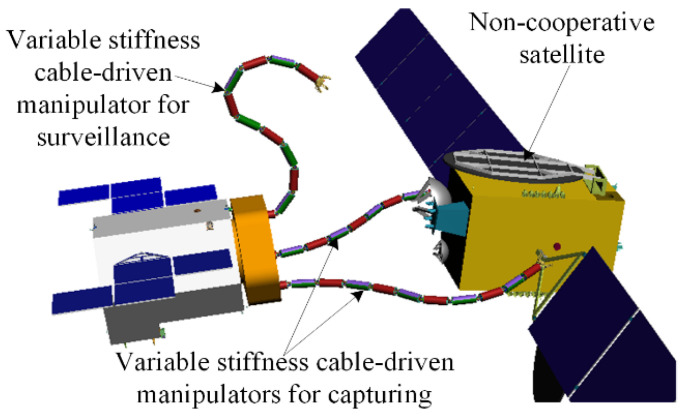
Concept of capturing a non-cooperative satellite.

**Figure 2 micromachines-13-02021-f002:**
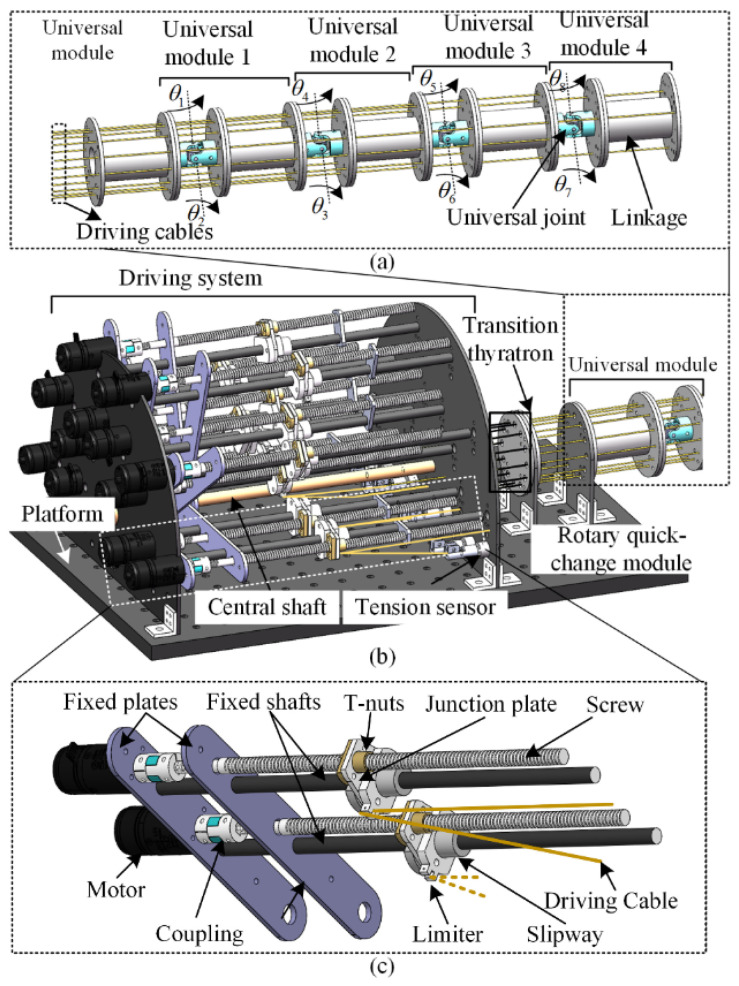
Structure of the cable-driven manipulator’s (**a**) universal module, (**b**) drive system, and (**c**) rotary quick-change module.

**Figure 3 micromachines-13-02021-f003:**
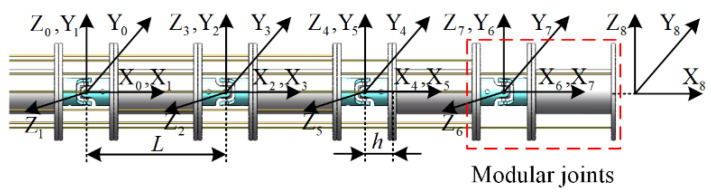
D-H coordinate system.

**Figure 4 micromachines-13-02021-f004:**
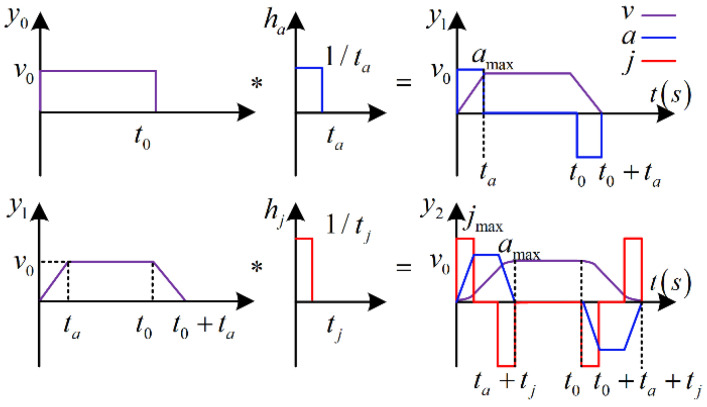
Principle of the convolutional dynamic-jerk-planning algorithm.

**Figure 5 micromachines-13-02021-f005:**
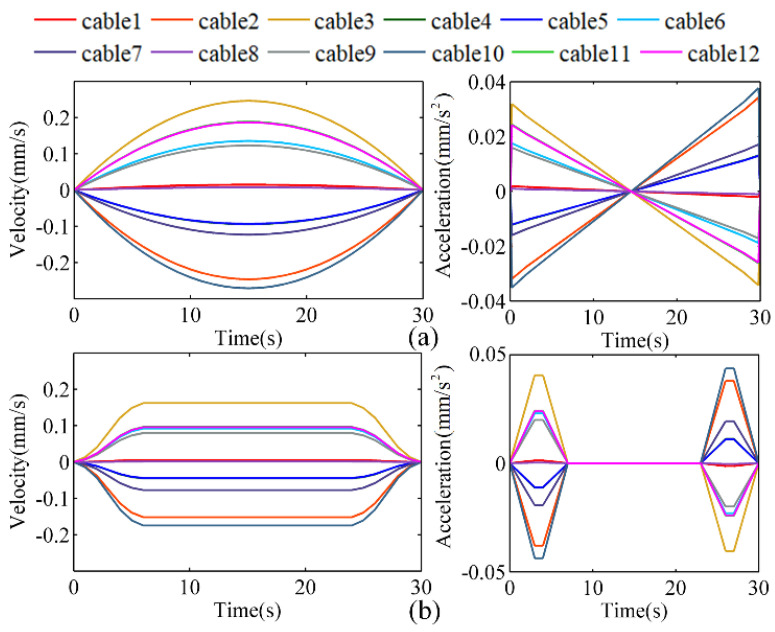
Velocity and acceleration curves of 12 cable−driven with different methods: (**a**) the iterative Jacobian method and (**b**) the convolutional dynamic-jerk-planning algorithm.

**Figure 6 micromachines-13-02021-f006:**
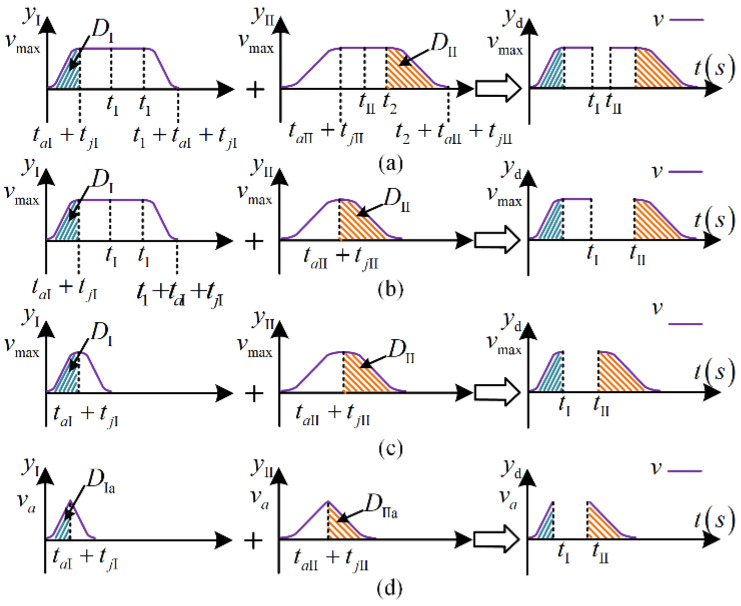
Convolutional dynamic-jerk-planning algorithm in different situations: (**a**) two phases of uniform motion, (**b**) one phase of uniform motion, (**c**) no uniform motion phases, and (**d**) no uniform acceleration phases.

**Figure 7 micromachines-13-02021-f007:**
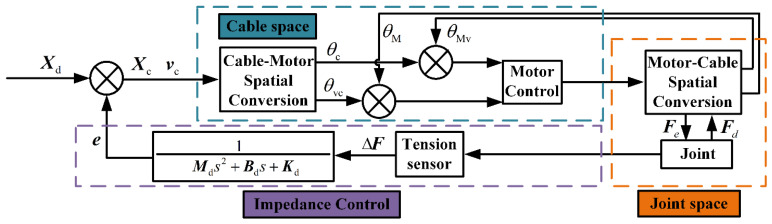
Position-based impedance control system.

**Figure 8 micromachines-13-02021-f008:**
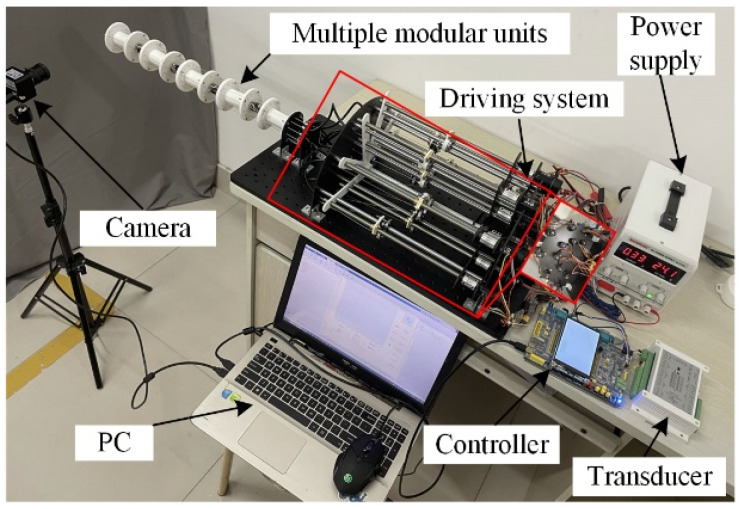
Variable-stiffness cable-driven manipulator system.

**Figure 9 micromachines-13-02021-f009:**
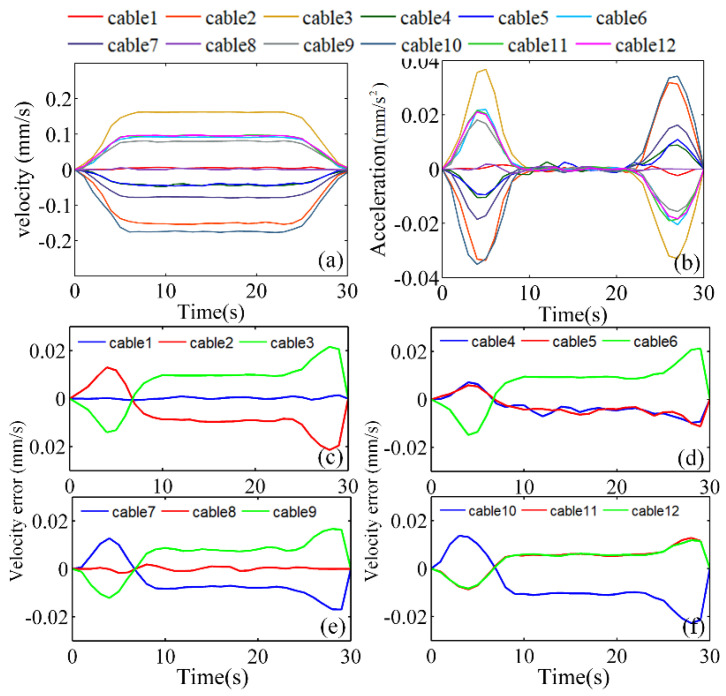
Velocity control of the 12 drive cables: (**a**) the actual velocity, (**b**) the actual acceleration, (**c**–**f**) the velocity errors of modules 1, 2, 3, and 4, respectively.

**Figure 10 micromachines-13-02021-f010:**
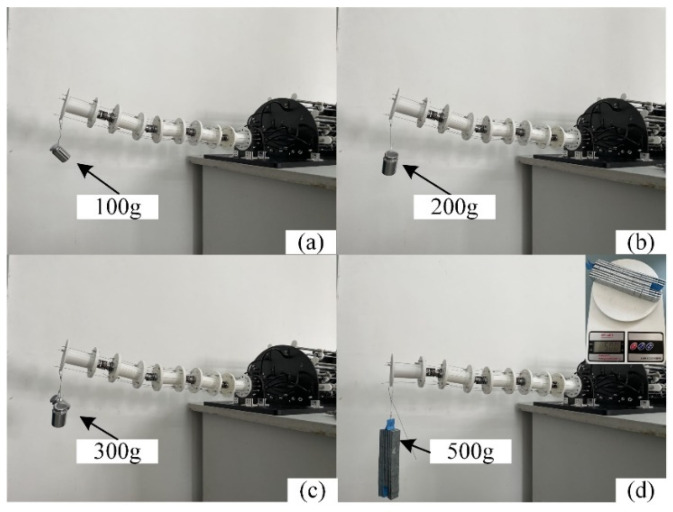
Load experiment (**a**) 100 g of weight, (**b**) 200 g of weight, (**c**) 300 g of weight, (**d**) 500 g of weight.

**Figure 11 micromachines-13-02021-f011:**
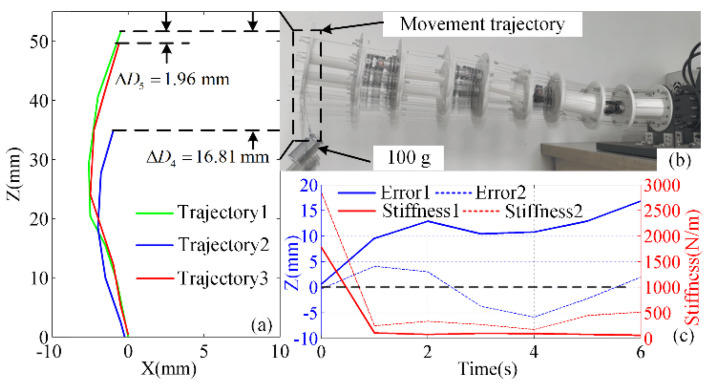
Dynamic stiffness experiments: (**a**) the displacement changes at the manipulator’s end for the three trajectories, (**b**) the manipulator’s full trajectories, and (**c**) the position errors of the manipulator’s end.

**Table 1 micromachines-13-02021-t001:** D-H Parameters.

Joint	$a_{i}\;\left(\mathrm{mm}\right)$	$\alpha_{i}\;\left({}^{\circ}\right)$	d (mm)	$\theta_{i}\;\left({}^{\circ}\right)$
1	0	90°	0	$\theta_{1}$
2	L	0°	0	$\theta_{2}$
3	0	−90°	0	$\theta_{3}$
4	L	0°	0	$\theta_{4}$
5	0	90°	0	$\theta_{5}$
6	L	0°	0	$\theta_{6}$
7	0	−90°	0	$\theta_{7}$
8	L	0°	0	$\theta_{8}$

## Data Availability

Not applicable.

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
