# Peer review of "A Convolutional Dynamic-Jerk-Planning Algorithm for Impedance Control of Variable-Stiffness Cable-Driven Manipulators"

_micromachines, 2022, doi:10.3390/mi13112021_

Round 1
Reviewer 1 Report
The paper proposes an impedance control algorithm and a convolution dynamic impact planning algorithm for variable-stiffness cable-driven manipulators. However, the experimental results and analysis are relatively simple. It is recommended to supplement experimental data that can be quantified and compared to fully illustrate the advantages of the proposed method. In addition, the figures in the paper are not standardized, and there are some statement errors, such as "Fig. 3" in line 183, "Where" in line 229, acceleration unit in Figure 9, and so on.
Author Response
请参阅附件。

Reviewer 2 Report
This paper proposes the convolutional dynamic-jerk-planning algorithm for impedance control of variable-stiffness cable-driven manipulators. First, a variable-stiffness cable-driven manipulator with universal modules and rotary quick-change modules is designed to overcome difficulties related to disassembly, installation, and maintenance. Second, a convolutional dynamic-jerk-planning algorithm is devised to overcome the discontinuity and shock problems of the manipulator’s velocity during intermittent control processes. The algorithm can also make acceleration smooth by setting jerk dynamically, reducing acceleration shock, and ensuring the stable movement of the cable-driven manipulator. Third, the stiffness of the cable-driven manipulator is further optimized by compensating for the position and velocity of drive cables by employing position-based impedance control.
The manuscript is well organized. The work is novel and interesting. My overall recommendation is to accept the paper. However, I have some comments that I feel would improve this paper. The following problems or concerns need to be addressed in the revision:
1. Please align fig and figure in the paper.
2. Section II. B: This section is not closely related to the topic of this paper and thus, it could be removed from the paper.
3. Page 8, the first paragraph: what is "over its long-term operation process"?
4. Delta_X defined in (34) is contradictory to "delta_x is the difference between X_d and X_double_dot".
5. One can read immediately after Figure. 5 "More motors are present in the cable-driven manipulator than in other manipulators." This is not necessarily true.
6. The unit of acceleration in Figure. 9 (b) is not correct. The vertical axes of Figure. 9 (c-f) should be velocity error, not velocity.
7. One can read from the paragraph immediately before Figure. 9 "The actual velocity and acceleration of the 12 drive cables are shown in Figure. 9 (a)." The actual velocity and acceleration of the 12 drive cables are actually shown in the Figures. 9 (a) and (b).
8. At the end of the paragraph immediately before Figure. 10, one can read "Stiffness 1 is the stiffness of trajectory 2." A trajectory does not have stiffness.
Reviewer 3 Report
The reviewer believe that the convolutional dynamic-jerk-planning algorithm is interesting to readers. However, there are still some issues that have not been clarified. My comments and questions are as follows:
1. The description of the developed cable-driven manipulator is not clear, especially the quick-change module. In Fig 2(c), two arrows, starting from “central shaft” and “fixed shaft”, respectively, are directed to the same part, a black-colored rod.
2. There exist mistake in elements of Eq (3).
3. The define of variable $\beta$ is confused. Actually, as per the design of the manipulator, $\beta$ should equal to $30 \degree$.
4. The type of cable should be given before the kinematic equation is derived. The paper only gives kinematic model of the first modular unit; however, the kinematic models of the other modular sets are relative to type of the cable employed in the design. Only if brake cable is employed, the models of the other units are similar to that of the first one, and each unit is decoupled, otherwise, not.
5. In the impedance control, the elastics of the cable is ignored. It might be reasonable for no-load case, but for grasping application, it might not. Please explain it.
6. In the test, the load of 100g weight seems to be too light, so that the practicability might be in doubt.
7. Mathematic writing should be improved.
Round 2
Reviewer 3 Report
The authors have mainly solved the concerns of the reviewers.